# Characterizing the Cell-Free Transcriptome in a Humanized Diffuse Large B-Cell Lymphoma Patient-Derived Tumor Xenograft Model for RNA-Based Liquid Biopsy in a Preclinical Setting

**DOI:** 10.3390/ijms25189982

**Published:** 2024-09-16

**Authors:** Philippe Decruyenaere, Willem Daneels, Annelien Morlion, Kimberly Verniers, Jasper Anckaert, Jan Tavernier, Fritz Offner, Jo Vandesompele

**Affiliations:** 1Department of Hematology, Ghent University Hospital, 9000 Ghent, Belgium; willem.daneels@uzgent.be (W.D.); fritz.offner@uzgent.be (F.O.); 2OncoRNALab, Cancer Research Institute Ghent (CRIG), Ghent University, 9000 Ghent, Belgium; annelien.morlion@ugent.be (A.M.); kimberly.verniers@ugent.be (K.V.); jasper.anckaert@ugent.be (J.A.); jo.vandesompele@ugent.be (J.V.); 3Department of Biomolecular Medicine, Ghent University, 9000 Ghent, Belgium; jan.tavernier@vib-ugent.be; 4Cancer Research Institute Ghent (CRIG), Ghent University, 9000 Ghent, Belgium; 5VIB-UGent Center for Medical Biotechnology, 9052 Ghent, Belgium; 6Orionis Biosciences B.V., 9052 Zwijnaarde, Belgium

**Keywords:** cell-free RNA, liquid biopsy, biomarkers, DLBCL, diffuse large B-cell lymphoma, PDTX, patient-derived tumor xenograft model

## Abstract

The potential of RNA-based liquid biopsy is increasingly being recognized in diffuse large B-cell lymphoma (DLBCL), the most common subtype of non-Hodgkin’s lymphoma. This study explores the cell-free transcriptome in a humanized DLBCL patient-derived tumor xenograft (PDTX) model. Blood plasma samples (n = 171) derived from a DLBCL PDTX model, including 27 humanized (HIS) PDTX, 8 HIS non-PDTX, and 21 non-HIS PDTX non-obese diabetic (NOD)-scid IL2Rgnull (NSG) mice were collected during humanization, xenografting, treatment, and sacrifice. The mice were treated with either rituximab, cyclophosphamide, doxorubicin, vincristine, and prednisone (R-CHOP), CD20-targeted human IFNα2-based AcTaferon combined with CHOP (huCD20-Fc-AFN-CHOP), or phosphate-buffered saline (PBS). RNA was extracted using the miRNeasy serum/plasma kit and sequenced on the NovaSeq 6000 platform. RNA sequencing data of the formalin-fixed paraffin-embedded (FFPE) tissue and blood plasma samples of the original patient were included. Flow cytometry was performed on immune cells isolated from whole blood, spleen, and bone marrow. Bulk deconvolution was performed using the Tabula Sapiens v1 basis matrix. Both R-CHOP and huCD20-Fc-AFN-CHOP were able to control tumor growth in most mice. Xenograft tumor volume was strongly associated with circulating tumor RNA (ctRNA) concentration (*p* < 0.001, R = 0.89), as well as with the number of detected human genes (*p* < 0.001, R = 0.79). Abundance analysis identified tumor-specific biomarkers that were dynamically tracked during tumor growth or treatment. An 8-gene signature demonstrated high accuracy for assessing therapy response (AUC 0.92). The tumoral gene detectability in the ctRNA of the PDTX-derived plasma was associated with RNA abundance levels in the patient’s tumor tissue and blood plasma (*p* < 0.001), confirming that tumoral gene abundance contributes to the cell-free RNA (cfRNA) profile. Decomposing the transcriptome, however, revealed high inter- and intra-mouse variability, which was lower in the HIS PDTX mice, indicating an impact of human engraftment on the stability and profile of cfRNA. Immunochemotherapy resulted in B cell depletion, and tumor clearance was reflected by a decrease in the fraction of human CD45+ cells. Lastly, bulk deconvolution provided complementary biological insights into the composition of the tumor and circulating immune system. In conclusion, the blood plasma-derived transcriptome serves as a biomarker source in a preclinical PDTX model, enables the assessment of biological pathways, and enhances the understanding of cfRNA dynamics.

## 1. Introduction

Diffuse large B-cell lymphoma (DLBCL) is the most common histological subtype of non-Hodgkin’s lymphomas, accounting for approximately 25% of all newly diagnosed cases [1]. Although a sustained complete response (CR) is obtained in most patients with first-line R-CHOP immunochemotherapy (rituximab, cyclophosphamide, vincristine, doxorubicin, and prednisone), patients with refractory or relapsed (R/R) disease have a poor prognosis, despite second-line treatments [2].

Two decades ago, gene expression profiling discriminated between different cell-of-origin DLBCL subgroups, germinal center B cell-like, and activated B cell-like, which was clinically implemented through the use of surrogate immunohistochemistry algorithms [3,4]. Since then, DLBCL has increasingly been recognized as a highly heterogeneous disease with respect to morphology, genetics, and biological behavior. Despite significant progress, the cell-of-origin division cannot fully reflect the heterogeneity in DLBCL, as was illustrated by several phase III trials that failed to identify those patients that could benefit from adding targeted agents to R-CHOP based on these classifiers [5,6,7]. More recently, integrative analyses using whole-exome and transcriptome sequencing have pinpointed distinct drivers of different molecular subgroups and have produced novel molecular classifications with prognostic implications [8,9,10,11,12,13]. Prospective trials evaluating subtype-informed therapy based on these refined classifications are currently ongoing [14,15].

A liquid biopsy is the process of investigating tumor-derived cells or biomaterials like cell-free nucleic acids, metabolites, proteins, or extracellular vesicles (EVs) through biofluid sampling without the need for a tissue biopsy. Potential advantages include the non-invasive nature, the ability to reflect inter- and intra-tumoral heterogeneity, and the possibility of repeated measurements through longitudinal profiling [16]. In this regard, there has been increased interest in different forms of coding and non-coding cell-free RNA (cfRNA) that play crucial roles in intercellular communication and contribute to proliferation, malignant transformation, angiogenesis, priming of future metastatic niches, and immune response escape [17,18,19].

The origins and roles of cfRNAs are still not fully elucidated, but it has been demonstrated that cfRNAs can be actively secreted or released during various cellular events, including apoptosis, with their levels being influenced by the viability and source of the cells. Evidence indicates that cfRNA levels can fluctuate in response to conditions such as hypoxia, heightened cellular metabolism, and endothelial activation [20,21,22]. Given their inherent instability due to ribonuclease degradation, endogenous cfRNAs are protected through several mechanisms, such as encapsulation within EVs or by forming ribonucleoprotein complexes [17,23].

However, most cfRNA transcripts are not tumor-specific and their abundance in the plasma is influenced by many other variables that also impact healthy organs and immune cells. As the enrichment of transcripts in the plasma compared to the tumoral compartment, its function, as well as their reciprocal interaction, are poorly understood, the identification of robust, tumor-specific cfRNA biomarkers remains challenging. Models that allow partial control of the underlying variability, such as patient-derived tumor xenografts (PDTX), could provide valuable new insights. In PDTX models, surgically resected tumor samples are engrafted directly into immune-compromised mice. Genome-wide gene expression analysis studies have demonstrated that PDTX models maintain the activities of most key genes and complete pathways in primary tumors and can even reflect intratumoral heterogeneity [24,25,26]. Compared to tumor cell lines, they better recapitulate the complex biological architectures and characteristics of the original tumors by maintaining the molecular, genetic, and histological heterogeneity through limited serial in vivo passaging in mice. Moreover, human engraftment of PDTX models provides the additional advantage of incorporating a functional human immune system (HIS). Throughout the manuscript, the HIS prefix will be used to refer to these “humanized” mice, those bearing a human immune system. As these models provide an excellent platform to study stromal–tumor interactions and can effectively capture patients’ responses to therapies, they have been extensively used in preclinical drug evaluation, biomarker identification, and drug screening for personalized treatment [27,28,29,30,31]. Previous research using a similar PDTX model as in the current study has demonstrated the ability to identify promising, antitumoral effects of new compounds in lymphoma and leukemia, including immunocytokines with cell-specific activity (“AcTaferons”; AFN), as well as to study the role of the microenvironment in establishing proficient antitumor effects [32,33].

The goal of this study is to explore cfRNA in longitudinally collected blood plasma in a DLBCL PDTX model, both HIS and non-HIS, to identify tumor-derived signals, as well as to assess functional changes in both the murine and human transcriptome during disease course and treatment, thereby investigating the potential of RNA-based liquid biopsy in a preclinical setting.

## 2. Results

### 2.1. PDTX Growth, Therapy Response, and Toxicity

A total of 171 longitudinal blood plasma samples derived from 8 humanized-only (HIS non-PDTX), 21 xenografted-only (non-HIS PDTX), and 27 both humanized and xenografted (HIS PDTX) NSG mice were included. The xenografted groups were randomized in three treatment arms: R-CHOP, huCD20-Fc-AFN-CHOP, and PBS (Figure 1).

Both R-CHOP and huCD20-Fc-AFN-CHOP were able to control tumor growth in the majority of mice (Figure 2). As the dose of rituximab in the former group was higher (250 µg) compared to the dose of huCD20-Fc-AFN (40 µg) in the latter group, this could explain a more effective control of tumor growth in the HIS mice due to a longer circulation of the active compound. No tumor growth was observed in 3 out of 10 PBS-treated HIS mice, indicating the potential of the humanized immune system to effectively control the tumors. All PBS-treated non-HIS mice showed tumor growth, albeit delayed in 2 out of 7 mice, indicative of the degree of heterogeneity within the PDTX model. The immediate toxicity was manageable but more pronounced in the HIS setting, as these mice had been exposed to non-lethal radiation prior to humanization. Nonetheless, all mice recovered. However, starting at 30 days following CHOP administration, significant late toxicities were observed, which we attributed to opportunistic infections as no clear systemic tumors were found at necropsy (Appendix A).

### 2.2. Blood Plasma Cell-Free RNA Concentration

Figure 3 depicts the evolution of cfRNA concentration after human engraftment, xenograft implantation, and treatment for both murine and human transcripts separately. Within the HIS mice, a low concentration of human cfRNA was present prior to xenograft implantation as a result of humanization (mean 3.50 ± 2.76 pg/mL). This concentration did not differ significantly from the PBS-treated HIS mice that were never xenografted (6.63 ± 6.04 pg/mL).

A clear increase in human cfRNA concentration was noted in the PBS arm, both in the HIS (prePDTX versus post-treatment *p* = 0.015; pre-treatment versus post-treatment *p* = 0.0070) and non-HIS mice (pre-treatment versus post-treatment *p* = 0.038; pre-treatment versus sacrifice *p* = 0.0023), suggesting that it derives from the growing PDTX and reflects circulating tumor RNA (ctRNA). This represented a 25.79-fold and an 18.61-fold increase in human cfRNA concentration during tumor growth in the PBS-treated group for HIS and non-HIS mice, respectively (Figure 3; Appendix A). In the non-HIS PBS-treated mice, in which the xenograft was the only source of human tumor-derived cfRNA (ctRNA), tumor volume was strongly associated with ctRNA concentration (*p* < 0.001, R = 0.89), with the fraction of ctRNA on total cfRNA (*p* < 0.001, R = 0.77), as well as with the number of uniquely detected human genes (*p* < 0.001, R = 0.79) (Figure 4; Appendix A). Overall, the human cfRNA concentration at late-stage xenograft growth was 15.54 and 10.10 times higher in the HIS PDTX mice (mean 103.27 ± 136.17 pg/mL) compared to both the HIS non-PDTX mice (*p* = 0.013) and the non-HIS PDTX mice (*p* = 0.039), respectively, indicating that both the humanization and xenografting contributed synergistically to the human cfRNA concentration (Appendix A). Similarly, the human cfRNA fraction was 16.69 and 10.15 times higher in the HIS PDTX mice (mean 3.05 ± 3.21%) compared to the HIS non-PDTX (*p* = 0.0018) and the non-HIS PDTX mice (*p* = 0.0084), respectively (Appendix A).

Lastly, both in HIS (in the R-CHOP arm) and non-HIS mice (all treatment arms), there was a significant increase in murine cfRNA. In the PBS-treated mice, this might be explained by the effects of tumor growth and invasion in murine organs. In the immunochemotherapy-treated mice, therapy-induced destruction of murine tissue and immune cells might induce increased cfRNA levels (Appendix A). Notably, tumor volume was also associated with an increase in murine cfRNA but not with the number of uniquely expressed genes (Figure 4 and Appendix A).

### 2.3. Differential Abundance Analysis

#### 2.3.1. Circulating Tumor cfRNA Profile

To demonstrate the presence of circulating tumor cfRNA in PDTX-engrafted mice, the ctRNA profiles of non-HIS mice with advanced lymphoma growth were compared between PBS treatment and immunochemotherapy. As visualized in Figure 5, the PBS-treated samples cluster based on the human, not the murine, gene abundance. The two PBS-treated outliers in the human counts (symbolized with ✝) were identified as the mice with the highest tumor volume at sacrifice that also had the highest human-to-total counts percentage (0.90% and 0.62%, respectively) and had amongst the highest human cfRNA concentrations (16 pg/mL and 18 pg/mL, respectively), underlining the correlation between tumor burden and ctRNA concentration. A total of 39 human differentially abundant genes (DAGs) were identified, all higher in the PBS-treated group (Appendix A). The majority of DAGs were coding (79.49%), but other classes included long non-coding RNAs (lncRNA), antisense RNAs, small Cajal body-specific RNAs (scaRNA), and small nuclear RNAs (snRNA). The top five abundant genes were *RNVU1-19*, *RELN*, *ACTB*, *TTN*, and *PTMA*. GSEA showed positive enrichment for MYC targets, allograft rejection, G2M checkpoint, and apical junction in the PBS-treated samples. To further illustrate the evolution in ctRNA during xenograft growth, a longitudinal impulseDE2 analysis was performed, showing a progressive increase in *RNUV1-19*, *RELN*, *ACTB*, and *PELP1* in the PBS-treated group due to xenograft growth as opposed to the treated group (Appendix A).

Concerning the HIS mice, the PBS-treated PDTX mice were compared to both immunochemotherapy-treated PDTX mice and PBS-treated mice without PDTX. Clustering could still be appreciated between the PBS-treated PDTX mice on the one hand and the other mice on the other hand (Figure 5). The samples that showed tumor growth despite immunochemotherapy (symbolized with *) converged toward the PBS-treated samples, again pinpointing that PDTX growth is the main driver of variance in the cfRNA profile. A total of 312 and 37 DAGs were identified when comparing PBS-treated PDTX mice to immunochemotherapy-treated PDTX mice and PBS-treated non-PDTX mice, respectively (Appendix A). As also expected here, all DAGs were highly abundant in the PBS-treated PDTX mice. The majority of DAGs were again coding (76.92% and 59.46%, respectively), and other classes included antisense RNAs, lncRNAs, snRNAs, mitochondrial ribosomal RNAs (mt-rRNA), microRNAs (miRNA), small nucleolar RNAs (snoRNA), and pseudogenes. GSEA showed enrichment for MYC targets, Wnt/β-catenin signaling, PI3K/AKT/mTOR signaling, mitotic spindle, G2M checkpoint, allograft rejection, interferon-gamma response, reactive oxygen species signaling, and apoptosis (PBS- versus immunochemotherapy-treated), and for MYC targets, Wnt/β-catenin signaling, G2M checkpoint, unfolded protein response, and PI3K/AKT/mTOR signaling (PBS-treated PDTX versus PBS-treated non-PDTX), respectively. The increased number of DAGs identified in the former comparison may indicate that especially the interplay of the (human) immune system and the xenograft acts as a major driver of cell-free transcriptional changes, compared to each separately. The DAGs between PBS- and chemotherapy-treated mice were linked to multiple enriched ontologies and pathways associated with malignant diseases (Appendix A). Moreover, DAGs were identified involved in the Golgi apparatus (*CNN2*, *RPGR*, *GOLGA6L6*, *GOLGA6L22*, *GOLGA6L10*, *GOLGA6L7*, *GOLGA6L3 pseudogene*), Toll-like receptor signaling (*PALM3*), mRNA splicing (*RBM25*, *CWC25*, *SF3A1*, *HELLPAR)*, apoptosis *(MDM4*; *AP5M1)*, regulation of NFkB pathway (*LIMD1*), Wnt–β-catenin–STAT3 signaling (*MYH9*), aerobic glycolysis (*PKM*), and methylation (*KMT2D*, *METTL7A*, *EIF3A*, *VIRMA)*, some of which have been associated with progression and prognosis in DLBCL [34,35,36,37,38,39].

Figure 6 depicts a Venn diagram of the overlap of DAGs between the different comparisons, illustrating an overlap of 8 human genes between all comparisons, which represents tumor-specific signals, irrespective of humanization: *PHF14*, *PELP1*, *RPL7L1*, *HPRT1*, *RELN*, *RNU1-75P*, *RNU5A-8P*, and *RNVU1-19*. A signature computed as the average expression value of these genes demonstrated high accuracy for assessing therapy response (AUC 0.92), indicating their tumoral origin. Six out of these eight genes were also differentially abundant in the FFPE tissue of a cohort of DLBCL patients, compared to healthy controls, and the gene signature allowed discrimination between both (AUC 0.84) (Appendix A). The gene signature was, however, not able to discriminate responders from non-responders to R-CHOP therapy based on its abundance in the diagnostic DLBCL FFPE samples (AUC of 0.60). Moreover, 7 out of these 8 genes were detectable in the blood plasma of the patient that provided the xenograft, with distinct expression trajectories from the time of diagnosis to progressive disease and complete response during and after treatment (Appendix A). RNVU1-19 was the only gene not found in both the patient’s FFPE tissue and blood plasma. Several genes (e.g., PHF14 or RELN) displayed an inverse abundance pattern in the blood plasma during treatment, as would be expected based on the abundance in the FFPE tissue. Overall, genes detected in the ctRNA of PBS-treated non-HIS PDTX mice were more abundant in the patient’s FFPE tissue and blood plasma, compared to the undetected genes in the ctRNA (two-sample Kolmogorov–Smirnov test, *p* < 0.001; Appendix A). Moreover, there was a moderate correlation between the abundance of a gene within the patient’s matched FFPE tissue and plasma sample (*p* < 0.001; R = 0.59) (Appendix A).

Lastly, the highest number of murine DAGs were found when comparing PBS- to immunochemotherapy-treated non-HIS PDTX mice. This finding suggests that the immune system (in this case, the murine) and its interaction with the xenograft during treatment drive transcriptomic changes in the plasma cfRNA profile.

#### 2.3.2. Variability in cfRNA Repertoire

Figure 7 shows the overlap of the human genes detected between the individual mice within the HIS non-PDTX, non-HIS PDTX, and HIS PDTX groups. In the non-HIS PDTX mice, *MED12L* and *TTN* were present in all the mice (n = 7), and *ACTB*, *RPL37*, *RNU1-75P*, and *RNVU1-19* were present in at least 6 out of the 7 mice. A total of 21.15% of human genes (690/3263) were detected in at least two different mice. When evaluating the ctRNA profile in matched early- and late-stage samples during PDTX growth, this overlap remained stable, with 23.32% (970/4160) of human genes detected in at least two different mice. Moreover, a significant, although small, overlap between the matched early- and late-stage samples of individual mice was demonstrated (*p* < 0.001; Jaccard index (JI) between 0.030 and 0.11; median 7.87%) (Appendix A). In the HIS non-PDTX mice (n = 8), only *RNVU1-19* was detected in all mice, and *RNU1-75P* and *TTN* were additionally detected in 7 out of 8 mice. A total of 20.22% (619/3062) of unique genes were shared by at least two mice. Lastly, the HIS PDTX mice (n = 8) shared a total of 23 human genes among all mice (*ROCK1*, *ACTB*, *RPL3*, *MYH9*, *RPL28*, *RPL19*, *HSPA8*, *RBM25*, *RPL5*, *AHNAK*, *HNRNPA1*, *RPS24*, *RPL7L1*, *LARP1*, *TTN*, *EEF1A*, *YWHAZ*, *PTMA*, *HIST1H1C*, *RNU1-75P*, *RF00019*, *TMSB4X*, *RNVU1-19*), and 40.91% (4779/11682) of unique genes were shared by at least two mice. There was a significant overlap between the matched early- and late-stage samples (*p* < 0.001; Jaccard index (JI) between 0.041 and 0.21; median 12.81%), overall larger than in the non-HIS PDTX mice (Appendix A). Lastly, in contrast to human counts, the vast majority of murine genes were detected in all mice within each group (Appendix A), illustrating that inter-murine variability is much smaller and that most murine genes in the plasma compartment are shared, irrespectively of engraftment or xenograft implantation.

### 2.4. Circulating Immune Cell Repertoire

#### 2.4.1. Flow Cytometry

To investigate the composition of the engrafted human immune system and its dynamic changes during treatment, flow cytometry was performed in the HIS mice (Appendix A). Although all mice were humanized simultaneously, using the same pool of umbilical cord blood, the immune cell composition was heterogeneous. Overall, huCD19+ B cells were the initial predominantly circulating immune component, evolving from naïve B cells to memory B cells. Subsequently, an increase in the proportion of T cells was noted, predominantly huCD4+ cells with a huCD4+/huCD8+ ratio ranging from 2.4 to 10.4 prior to tumor inoculation and from 2.6 to 10.8 at sacrifice. The majority of both huCD4+ and huCD8+ T cells in the blood compartment were of the memory phenotype, and the proportion of naive T cells was the lowest at sacrifice. In the PBS-treated HIS mice, tumor growth coincided with an increase in B cells with a memory phenotype, presumably reflecting circulating tumor cells (*p* = 0.024; 95% CI [0.0032%, −0.24%] for pre-PDTX versus sacrifice timepoint). As expected, treatment with either R-CHOP or huCD20-fc-AFN-CHOP significantly impacted the circulating immune cell repertoire, with a strong B cell depletion (*p* = 0.0013; 95% CI [−73.61%, −25.53%] for pre-treatment versus sacrifice), leading to a proportional increase in the T cell fraction (*p* = 0.0037; 95% CI [16.69%, 53.85%] for pre-treatment versus sacrifice), with especially an increase in huCD8+ T cells (*p* = 0.0032; 95% CI [5.10%, 27.75%] for pre-treatment versus sacrifice). Successful tumor clearance was also reflected by a decrease in huCD45+ on hu/muCD45+ cell ratio (*p* = 0.016 95% CI [−16.58%,−2.86%] for pre-treatment versus post-treatment) in the immunochemotherapy-treated group. This ratio was also significantly lower compared to the PBS-treated PDTX mice at the post-treatment timepoint (*p* = 0.0017; 95% CI [−58.41%,−11.38%]), which can be attributed to the decrease in the fraction of huCD19+ cells (*p* = 0.03; 95% CI [−53.66,−2.20]) in the former compared to the latter. There were no significant differences in huCD4+ T cell subsets (huCD4+ memory, huCD4+ naive, huCD4+ Treg, huCD4+ Tfh). Lastly, although there was a trend of increasing NK cells and decreasing myeloid cells in the immunochemotherapy-treated mice, significance was not reached. No differences could be noted between both treatment groups (R-CHOP versus huCD20-fc-AFN-CHOP) (Appendix A).

At the sacrifice timepoint, a significantly higher huCD45+ on hu/muCD45+ cell ratio was demonstrated in both the bone marrow and spleen of PBS-treated PDTX mice compared to the immunochemotherapy-treated mice (*p* = 0.012; 95% CI [−62.44%,−19.30%] for bone marrow and *p* < 0.001; 95% CI [−76.48%, −23.91%] for spleen) (Appendix A). R-CHOP- or huCD20-Fc-AFN-CHOP-treated mice showed a significant reduction of splenic B cells, more specifically of memory phenotype (*p* < 0.001; 95% CI [−63.95, −16.68%] for R-CHOP and *p* < 0.005; 95% CI [−54.79, −6.36%] for huCD20-Fc-AFN-CHOP). No differences in total huCD4+ or huCD8+ T cells were observed in the bone marrow or spleen upon treatment. When comparing R-CHOP to huCD20-fc-AFN-CHOP-treated PDTX mice, however, an increased fraction of naive huCD8+ cells was noted in the spleen (*p* = 0.041; 95% CI [0.5498, 46.16%]), not present in the blood or bone marrow.

#### 2.4.2. Computational Deconvolution Using cfRNA

In a complementary approach, the circulating immune compartment was investigated by applying computational deconvolution to the human cfRNA profile. Figure 8 depicts the relative contribution of immune cells over the course of xenograft implantation and treatment. Similar to the results obtained by flow cytometry, a decrease in B cells was noted in the HIS mice after R-CHOP or huCD20-fc-AFN-CHOP treatment (*p* = 0.039), not present in the PBS-treated mice. However, in the PBS-treated non-HIS mice, a clear increase in plasma cell contribution was noted after engraftment until sacrifice, directly related to tumor growth (*p* = 0.0015 pre-treatment versus sacrifice; 95% CI [11.27%–63.89%] for the non-HIS PDTX mice). This is in line with the differential abundance analysis, in which a significant proportion of the DAGs (10 out of 39 DAG; 25.64%) identified between PBS- and immunochemotherapy-treated non-HIS mice, were part of a plasma cell signature, including *IGHM*, *SSR3*, and *CD74* (Vorperian et al. [40]). When comparing the deconvolution results to those of the FFPE and plasma sample of the patient from which the PDTX is derived, a plasma cell fraction is noted within the diagnostic FFPE tissue but not in the blood plasma sample (Appendix A). Other relative changes in immune cell types during PDTX growth included an increase in mature conventional dendritic cells (*p* = 0.01; 95% CI [8.91%, 23.01%] for pre-treatment versus sacrifice) and monocytes (*p* = 0.033; 95% CI [1.06%, 8.24%] for pre-treatment versus sacrifice), as well as a decrease in macrophages (*p* = 0.036; 95% CI [−30.75%,−0.0033%] for pre-treatment versus sacrifice).

Lastly, the flow cytometric and computational deconvolution results on blood plasma were correlated. For NK cells (r = 0.22; *p* = 0.025) and T cells (r = 0.2; *p* = 0.043), there was a significant but weak correlation between both methods. For B cells (r = 0.094; *p* = 0.35) and myeloid cells (r = 0.11; *p* = 0.26), no significant correlation could be demonstrated.

## 3. Discussion

Cell-free nucleic acids have increasingly been recognized as valuable precision medicine biomarkers in cancer research, including lymphoproliferative malignancies. Although the function and origin of cfRNA remain largely unexplored, both human and xenograft studies have illustrated its potential to reflect intra- and inter-tumoral heterogeneity, as well as functional changes during disease course and treatment [41,42,43]. Total RNA sequencing, in particular, offers a hypothesis-independent, unbiased approach to robustly quantify gene abundance alterations of the entire cell-free transcriptome [42,44,45,46]. Moreover, by distinguishing human from murine sequencing reads in fragmented and low abundant RNA present in murine plasma by a highly accurate combined mapping strategy, the impact of humanization, xenografting, and their interplay can be investigated separately. We present the first study to longitudinally investigate cfRNA concentration and abundance in a humanized DLBCL PDTX model and to explore its potential as a source of tumor-specific biomarkers, as a tool for response assessment, as well as elucidation of underlying biological processes.

Higher ctDNA levels have been observed in DLBCL patients compared to healthy controls [47,48]. Similarly, we recently reported increased cfRNA blood plasma concentrations in a DLBCL patient cohort compared to healthy controls, with decreasing levels during successful first-line R-CHOP therapy [49]. Our results in the current study confirm these findings in an orthotopic DLBCL PDTX model and demonstrate that the ctRNA concentration in the blood plasma is strongly correlated to tumor growth and volume, illustrating that circulating RNA levels by itself may reflect tumor burden and therapy response. A correlation between ctRNA concentration and tumor volume, both for short and for long cfRNA subtypes, has been reported in the setting of neuroblastoma PDTX models but not yet in lymphoma [50,51,52]. Moreover, both humanization and xenografting contributed to the cfRNA concentration, but especially their combined presence and interaction entail a major source of cfRNA transcripts (more than tenfold increase when combined versus each separately).

Differential abundance analysis was able to discern tumor-bearing from successfully treated mice and to reveal transcripts with biomarker potential. In a murine Smurf2-deficient model of spontaneous DLBCL initiation, the presence of distinct miRNAs in the serum, long before actual tumor formation, was identified, the majority of which were also found enriched in PDTX DLBCL models [53,54]. In addition to miRNAs, our results illustrate the presence of other RNA subclasses, including mRNA, lncRNA, sn(o)RNAs, and pseudogenes. The majority of differentially abundant genes (DAGs) were coding, in a similar range as reported in solid tumor PDTX models [43]. Moreover, several markers changed in abundance during disease course and treatment, representing markers with the potential to complement PET-CT imaging in response assessment or to improve early relapse detection. In addition to cfRNA concentration, humanization also increased the number of unique DAGs between PBS- and immunochemotherapy-treated mice, indicating that the interplay of the immune system and the xenograft not only contributes to the abundance but also to the spectrum of cfRNA. Similarly, in the HIS mice, DAGs were identified within a wider range of RNA biotypes.

We pinpointed 8 human-derived genes (*PHF14*, *PELP1*, *RPL7L1*, *HPRT1*, *RELN*, *RNU1-75P*, *RNU5A-8P*, *RNVU1-19*), likely to represent tumor-specific signals as a signature combining these genes demonstrated high accuracy in assessing therapy response. The majority of these genes were also found differentially abundant in the diagnostic FFPE tissue of a cohort of DLBCL patients compared to healthy lymph node FFPE tissue, and the signature allowed discrimination between both. However, the signature was unable to discriminate responders from non-responders to first-line R-CHOP treatment based on its abundance in this cohort of DLBCL patients. This indicates that, while the signature reflects tumor-derived signals in both the PDTX and human settings, it does not allow upfront prediction of therapy response within this patient cohort. This limitation presumably arises because a single PDTX model cannot capture the extensive heterogeneity of the biology and tumor resistance mechanisms in DLBCL.

When comparing the abundance profile of the patient’s plasma during the disease course, inverse patterns are shown in a subset of the genes within the signature, contrary to what would be expected when compared to the abundance levels in the patient’s FFPE sample and the PDX plasma profiles of tumor-bearing mice. This observation has already been made in multiple human studies that have shown discrepancies between the plasma- and tissue-derived cfRNA repertoire for both coding and non-coding RNAs, in which no correlation, or even an opposite abundance profile, was demonstrated [49,55,56]. More specifically, in a recent study, approximately half of the DAGs identified between the FFPE tissue samples of DLBCL patients and healthy controls showed a different direction of dysregulation in the blood plasma, underlining that the latter has a unique transcriptomic profile with its own dynamics [49]. Several causes may underlie this finding. First, variability in (vesicle-mediated) secretion or passive release of RNA molecules in the bloodstream may result in different abundance patterns between both compartments. Second, although the transcripts have been shown to be clearly expressed by the xenograft, it is possible that they are also shed from other tissues besides the tumor compartment, e.g., by healthy organs or the patient’s immune cells due to the influence of tumor invasion or immunochemotherapy. Lastly, the plasma stability of some transcripts between the human and murine settings could be impacted by the rate and manner of degradation due to differences in the concentration of (human) proteins, lipids, and regulatory RNAs, resulting in an altered half-life in the blood stream.

Overall, genes identified within the ctRNA of the PDTX samples were found enriched within the diagnostic FFPE tissue and blood plasma sample of the patient from whom the xenograft originated. Moreover, a significant correlation between the abundance of a gene within the patient’s matched FFPE tissue and plasma sample was noted. These results suggest that if a gene’s abundance is higher in the patient’s tumor, it has an increased chance of being more abundant in the blood plasma. This, however, does not exclude the possibility of selective enrichment within the ctRNA profile due to other influencing factors, such as encapsulation in EVs or platelets. In this context, previous studies in DLBCL patients have shown conflicting data on whether tumor-derived RNA was enriched in EVs compared to serum or plasma [57,58,59,60,61].

Decomposing the tumor-derived transcriptome revealed high inter- and intra-murine variability. Only a relatively small fraction of ctRNA transcripts were shared, as well as continuously expressed during tumor growth, illustrating that ctRNA, even in a maximally controlled setting in which the xenograft is the only source of human transcripts and without interfering treatments, is released dynamically in the bloodstream and influenced by many factors. The large variability in ctRNA abundance across individual mice in non-HIS PDTX models has been previously observed and attributed to genuine biological variation, even in the setting of large tumor masses [43,50]. Similarly, high plasma transcriptome variability was demonstrated in various solid and hematological cancers [62]. This observation is not unique to ctRNA, as high variability has also been demonstrated for circulating tumor cells (CTCs), whose numbers in blood are, for example, strongly impacted by the timing of the draw due to the short half-life [63]. In the HIS PDTX mice, however, both the fraction of shared transcripts between mice and within each mouse over time were approximately twice as high. This indicates that the interplay between the human immune system and the xenograft not only significantly contributed to the RNA concentration and spectrum but also resulted in reduced variability, traits that are highly relevant in the exploitation of cfRNA as a robust biomarker source. Our findings, therefore, not only recommend the use of HIS engraftment in preclinical PDTX liquid biopsy research but also advocate for caution when interpreting cfRNA dynamics during follow-up or treatment in clinical studies, a setting in which the immune system is often impacted to different degrees, depending on the therapy.

Multiple recent studies have evaluated the effect of new (combinations of) compounds in DLBCL PDTX models, including their modulation of the tumor microenvironment [33,64,65,66]. Our flow cytometric analysis on whole blood samples of PBS-treated HIS mice showed that tumor growth coincided with an increase in B cells with a memory phenotype, presumably reflecting circulating tumor cells. Treatment with either R-CHOP or huCD20-Fc-AFN-CHOP resulted in a strong B cell depletion and successful tumor clearance was also reflected by a decrease in the fraction of human CD45+ cells. A similar pattern was observed in the spleen and bone marrow as treatment with immunochemotherapy induced depletion of splenic (malignant) B cells with a memory phenotype. Lastly, an increased fraction of naive cytotoxic T cells was noted in the spleen of the former group. The early involvement of bone marrow and spleen has been previously reported in a DLBCL PDTX model, in which a miRNA signature was also expressed in the spleen and bone marrow in DLBCL-forming mice starting at 3 months of age, not present in the control mice [53].

Similar to flow cytometry, a reduction in the B cell fraction was noted in the HIS mice after immunochemotherapy treatment in the cfRNA bulk deconvolution data. This is in line with our previous study, in which a clear decrease in cell-free mRNA of B cell-related markers in the blood plasma of R-CHOP responders was demonstrated as opposed to non-responders, reflecting a good response to first-line therapy [49]. Moreover, in the PBS-treated mice, a relative increase in plasma cell contribution was noted, directly related to tumor growth. The PDTX was derived from a patient with a non-GC DLBCL with plasmacytoid differentiation, suspected to have transformed from a previously undiagnosed marginal zone lymphoma, which could explain this finding. As this plasma cell contribution was detected in the deconvolution results of the diagnostic FFPE tissue sample but not in the blood plasma sample of the patient from which the PDTX was derived, this might indicate that more subtle tumor-derived signals can become difficult to detect in the blood plasma due to healthy tissue-derived contributions. In healthy humans, it has indeed been demonstrated that whole blood is the major contributor (~40%) to the cell-free transcriptome, with the bone marrow and lymph nodes also strongly contributing, with very high interpersonal variability in cfRNA [67,68,69]. The dissection of bulk cfRNA in PDTX models could, therefore, represent a valuable, complementary tool to capture underlying biological changes in the tumor and immune system in response to therapy in a preclinical setting, as well as enhance our understanding of the origin of cfRNA in the human liquid biopsy research field.

Our study has several limitations. First, the results were obtained from low-volume plasma samples (80 µL plasma). Evaluating a larger animal model, such as a xenografted rat, may provide additional insights into the biomarker potential of liquid biopsies of xenografts as more plasma can be prepared for additional analyses. Second, the impact of pre-analytical variables on cfRNA abundance results has been increasingly reported [70,71]. Therefore, direct comparison with other studies using different preanalytics (e.g., blood collection tube, centrifugation protocol, RNA extraction method) is challenging. All samples in this study were uniformly processed and, based on previous research showing that ctRNA does not primarily reside in murine or human platelets, platelet-free plasma was chosen as it displays a greater tumor gene diversity [43,72]. Moreover, EDTA tubes were used as these have demonstrated favorable results in cfRNA benchmarking research [70]. Third, as this study examined the circulating transcriptome in blood plasma, future research is needed to establish differential enrichment in other biofluids. Moreover, in the human setting, endogenous circulating RNAs are protected by several mechanisms, such as encapsulation within EVs, or they form ribonucleoprotein complexes with RNA-binding proteins that protect them from nuclease activity [17]. Therefore, the selectivity and degradation rate of human cfRNA can differ in mouse plasma compared to human plasma and may impact its abundance profile. Lastly, although HIS models allow the evaluation of human immune responses to different therapies in a more preclinically relevant setting, there are cross-species differences between the engrafted human cells and the mouse microenvironment in which they reside. These differences may influence cell–cell interactions, hematopoietic cell homing, survival, and expansion [73].

In conclusion, our study demonstrated the feasibility of longitudinally investigating cfRNA in both a HIS and non-HIS PDTX DLBCL model using low-volume blood plasma samples. As there is a strong correlation between ctRNA concentration and tumor volume, and decreasing values were demonstrated in successfully treated mice, these results support the use of cfRNA concentration to assess tumor burden and therapy response. Using abundance analyses, biomarkers were pinpointed that allowed discrimination between tumor-bearing and tumor-free mice with an 8-gene signature demonstrating high accuracy value for assessing response to therapy. The tumoral gene detectability in the ctRNA of the PDTX-derived plasma was correlated with RNA abundance levels in the patient’s tumor tissue and blood plasma, indicating that tumoral gene abundance is a contributing factor to the cfRNA profile. However, our analyses also reconfirm the high inter- and intra-mouse variability in cfRNA, even in a maximally controlled setting. Human engraftment, however, reduced this variability, implying that the interaction between xenograft and the human immune system plays a role in both the spectrum and the stability of the cell-free transcriptome. Lastly, computational deconvolution on bulk cfRNA provided biological insights on tumor growth and therapy-induced immune response, complementary to flow cytometry. Overall, the findings in this study support the unexplored potential of cfRNA blood plasma liquid biopsies as a biomarker source, as well as a tool to functionally assess biological pathways in preclinical PDTX models and to unravel the origin of cfRNA.

## 4. Materials and Methods

### 4.1. Humanized PDTX Model and Treatments

Humanized NOD-*scid* IL2Rgnull (NSG) mice were generated as previously reported [32,33]. Briefly, newborn NSG pups were sublethally irradiated with 100 cGy and intrahepatically injected with 1 × 10^5^ pooled CD34+ hematopoietic stem cells (HSCs) isolated from human umbilical cord blood. Human immune system (HIS) reconstitution was established 6 to 8 weeks following human stem cell (HSC) injection. Subsequently, a patient-derived DLBCL tumor xenograft was injected orthotopically near the inguinal node at 13.5 weeks after humanization. The xenograft was derived from a 62-year-old female, non-GC, double expressor DLBCL, without *BCL2/MYC* rearrangement and CISH EBV (EBER) negative. As a kappa clonal population with plasmacytoid differentiation was described in the patient’s diagnostic DLBCL FFPE tissue sample, a transformation from an underlying, previously undiagnosed marginal zone lymphoma was suspected.

Mice were stratified based on sex and human engraftment levels into treatments with R-CHOP (n = 10), huCD20-Fc-AFN-CHOP (n = 10), or PBS (phosphate-buffered saline) (n = 10). Treatments were started at D9 post-tumor inoculation using individual, sterile 29G needles and administered 4 times 1x/week IV, except for CHOP, which was only administered once due to toxicity concerns. Rituximab (Mabthera, Roche, Rotkreuz, Switzerland) was dosed at 250 µg, huCD20-Fc-AFN at 40 µg and CHOP was composed of 75 µg doxorubicin (Adriblastina, Pfizer, New York, NY, USA), 620 µg cyclophosphamide (Endoxan, Baxter, Deerfield, IL, USA), 6.25 µg vincristine (Vincrisin, Teva, Tel Aviv, Israel) and 10 µg dexamethasone (Sigma, Virginia Beach, VA, USA). The huCD20-Fc-AFN consists of a huCD20-specific single-domain antibody (VHH) linked through a heterodimeric ‘knob-in-hole’ human IgG1 Fc molecule to an attenuated huIFNa2 sequence as recently described [33]. In parallel, we evaluated the same treatments (R-CHOP/huCD20-Fc-AFN-CHOP/PBS) in non-HIS NSG mice (n = 7/7/7) and in non-tumor bearing HIS (n = 8/8/8) and non-HIS (n = 7/7/8) NSG mice.

Mice were maintained in pathogen-free conditions in a temperature-controlled environment with 12/12 h light/dark cycles and received food and water ad libitum. Animal experiments followed the Federation of European Laboratory Animal Science Association (FELASA) guidelines and were approved by the Ethical Committee of Ghent University (EC number 19-10k/19-01). Mice were checked daily. Weighing and tumor measurement by calipers were performed 3 times a week. All mice were sacrificed in week 29 post-humanization or earlier because of humane endpoints such as tumor volume > 1500 mm^3^ (L*W^2^/2), significant organomegaly, weight loss >20%, or development of significant xenogeneic graft-versus-host-disease.

### 4.2. Sample Collection and Preparation

#### 4.2.1. Murine Samples

A total of 171 longitudinally collected plasma samples derived from 35 HIS and 21 non-HIS NSG mice were collected during the period March 2022 and August 2022 at the VIB-UGent Center for Medical Biotechnology (Ghent, Belgium). Longitudinal plasma samples were drawn before and after humanization, xenograft implantation, and treatment or during disease progression. Blood samples were collected from a tail vein puncture into Microvette 200 EDTA K3E tubes (#SARS20.1288; VWR) and processed into platelet-free plasma within 1 h after blood draw. Platelet-free plasma was obtained by using a two-step centrifugation protocol (2 × 2500× *g* 10 min without brake at room temperature) and subsequently frozen and stored at −80 °C. Figure 1 shows a graphical representation of the longitudinal plasma samples taken from each group.

#### 4.2.2. Human Samples

Total RNA sequencing data of diagnostic FFPE tissue samples derived from a cohort of 27 DLBCL NOS patients, as well as of 21 FFPE tissue samples derived from non-malignant lymph node tissue, were obtained from the European Genome-Phenome Archive (EGAD00001011679). These samples were collected during the period June 2016 and September 2021 at Ghent University Hospital in Ghent (Belgium) and AZ Delta hospital in Roeselare (Belgium). The characteristics of the DLBCL patients can be found in Appendix A. Lastly, blood plasma samples of the patient from whom the PDTX was derived were obtained from the European Genome-Phenome Archive (EGAD00001011679). The blood samples were drawn at the diagnostic timepoint (sample ID RNA020137), at progressive disease after 6 cycles of R-CHOP therapy (sample ID RNA020144), and at complete remission after 2 cycles of R-GDP and local radiotherapy (sample ID RNA020153). RNA extraction and sequencing were performed as previously described [49].

### 4.3. Flow Cytometric Analysis

Human immune cells were extracted from the spleen and bone marrow (femur) by mechanical dissociation. We used the buffy coat obtained after the first centrifugation for platelet-free plasma preparation from K3 EDTA collected blood from humanized mice. All flow cytometry reagents and cell staining buffer (#420201) were purchased from Biolegend (San Diego, CA, USA). Optimal concentrations were determined by titration. RBCs were lysed using an in-house RBC lysis buffer. Murine and human Fc receptors were blocked using purified Rat Anti-Mouse CD16/CD32 (#101302) and Human TruStain FcX (#422302). Zombie NIR (#423105) was used to determine viability. Surface staining was performed for 20 min in the dark at 4 °C using 2 separate panels; Panel 1: huCD56-BV421 (#362551), huCD19-BV510 (#302241), muCD45-BV650 (#103151), huCD45-FITC (#304005), huCD33-PE (#303403), huCD27-PE-Cy7 (#303403), huCD70-APC (#355109), huCD3-AlexaFluor700 (#344821) and Panel 2: huCD197-BV421 (#353207), huCD8-BV510 (#301047), huCD25-BV605 (#302631), muCD45-BV650 (#103151), huCD45-FITC (#304005), huCD185-PE (#356903), huCD4-PE-Cy5.5 (#300529), huCD45RA-PE-Cy7 (#304125), huCD127-APC (#351315), and huCD3-AlexaFluor700 (#344821). Samples were measured on the MACSQuant Analyzer 16 (Miltenyi Biotec, Bergisch Gladbach, Germany) and analyzed in FlowLogic 8.6 (Inivai Technologies, Mentone, VIC, Australia).

### 4.4. RNA Extraction

RNA extraction was performed on 80 µL of plasma using the miRNeasy serum/plasma kit. In each plasma sample, 2 µL of Sequin spike-in controls were added to the sample lysates (1/260,000 of stock solution mix A; Garvan Institute of Medical Research), as well as 2 µL of External RNA Control Consortium (ERCC) spike-in controls (1/200,000; ThermoFisher Scientific, 4456740) were added to the RNA eluate. Genomic DNA was removed by adding 1 μL HL-dsDNase (ArcticZymes, 70800-202, Tromso, Norway) and 1.6 µL reaction buffer (ArcticZymes, 66001) to 12 µL RNA eluate, followed by 10 min incubation at 37 °C, and 5 min incubation at 55 °C. RNA was stored at −80 °C and thawed on ice immediately before library preparation.

### 4.5. Library Preparation and Sequencing

Total RNA sequencing libraries were prepared starting from 8 µL of RNA eluate using the SMARTer Stranded Total RNA-Seq Kit v3—Pico Input Mammalian (Takara Bio Europe, Paris, France, #634487) according to the manufacturer’s protocol. Equimolar library pools were prepared based on qPCR quantification with the KAPA Library Quantification Kit (Roche Diagnostics, Brussels, Belgium, #KK4854). The libraries were paired-end sequenced (2 × 100 nucleotides) on a NovaSeq 6000 instrument using a NovaSeq S2 kit (Illumina, San Diego, CA, USA, 20028315) with standard workflow loading of 0.65 nM (2% PhiX). BCL files generated by the Illumina sequencing system were processed using the Illumina Bcl2fastq (v.2.20) software to generate and demultiplex fastq files. Raw reads were assessed for quality using FastQC (v.0.11.9) [74]. To differentiate the circulating human transcriptome from the murine extracellular RNAs, a host–xenograft deconvolution algorithm for RNA sequencing read analysis was applied as previously described (https://github.com/CBIGR/exRNAxeno (accessed on 10 January 2024)) [43]. In short, reads were mapped using STAR (v.2.6.0) to a combined human and murine reference genome using the Ensembl GRCh38.94 (human) and GRCm38.94 (murine), respectively. Uniquely mapped reads were selected based on the NH:i:1 tag (SAMtools v.1.8, Pysam) [75]. BAM files were further masked by intersectBed for regions where control murine and human liquid biopsies empirically showed misalignment to the other reference genome (SAMtools v.1.8, BEDtools v.2.27.1, BEDOPS v.2.4.32) [75,76,77]. Further quality control on the filtered BAM files was performed using MultiQC (v.1.7), SAMtools (v.1.8), RseQC (v.2.6.4), and BEDTools (v.2.27.1) [75,76,78,79]. Finally, read counts of name-sorted BAM files were generated by HTSeq-count (v.0.11.0) [80].

### 4.6. Cell-Free RNA Concentration

RNA concentration was determined as previously described [81]. Briefly, the mass of a top abundant spike-in control (ERCC00130) was calculated based on the input concentration and volume of spike-in mix added to the sample. The corresponding RNA concentration was then estimated by multiplying the ERCC00130 spike mass by the ratio of total reads mapped to the endogenous human genome and the number of reads mapped to the specific ERCC00130 spike, and finally dividing the obtained mass by the plasma volume of the sample.

### 4.7. Differential Abundance and Principal Component Analysis

Differential expression analysis on normalized counts was performed using DESeq2 (v.1.36.0) [82]. In the DESeq2 result table, genes with a Benjamini–Hochberg corrected *p*-value (*q*-value) below 0.05 were considered differentially abundant. Volcano plots were visualized using EnhancedVolcano (v.1.14.0), and Venn diagrams were made using VennDiagramm (v.1.7.3) [83,84]. Concerning the longitudinal differential abundance analysis, the ImpulseDE2 algorithm (v.1.6.1) was applied using a *q*-value below 0.05. ImpulseDE2 models the gene-wise expression trajectories over time with a descriptive single-pulse function, which is based on a negative binomial noise model with dispersion trend smoothing by DESeq2 [85]. Principal component analysis using the package pcaExplorer (v.3.19) was performed to reduce the dimensionality of the gene expression data and to visualize the relationships between samples based on their overall gene expression profiles. This tool enables the visualization of samples on a two-dimensional plot based on the first two principal components (PC1 and PC2), which capture the largest variance in the data. The loadings of individual genes on the principal components were also examined, indicating their contribution to the respective principal components, hereby highlighting the key genes that drive the observed differences between the samples in the PCA plot [86].

### 4.8. Gene Set Enrichment Analysis

Gene set enrichment analysis using GSEA (v.4.2.3) was performed to explore the functionally enriched pathways and hallmark gene sets related to subgroups [87]. A pre-ranked gene GSEA was performed based on log2 transformed fold changes between different groups obtained from DESeq2 differential abundance analysis. Hallmark and Canonical Pathways gene sets were obtained via the Molecular Signatures Database MSigDB (v.7.5.1) [88]. Pathways were up- or down-regulated according to the enrichment score (ES) that represents the degree to which a set was over-represented at the top or bottom of the ranked list, respectively. To explore Gene Ontology (GO), the enrichGO() function from the R package clusterProfiler (v.4.0.5) was used [89]. The GO annotation file for the human species was obtained from the Gene Ontology (available online at: http://geneontology.org/ (accessed on 1 December 2023)) [90,91]. Enrichment analysis based on the Network of Cancer Genes (NCG) database (http://ncg.kcl.ac.uk/index.php (accessed on 1 December 2023)) was performed using the enrichNCG() function [92]. An enrichpathway function of ReactomePA (v.1.34.0) was used to perform pathway analysis using the Reactome pathway database (v.1.74.0) [93]. Enrichplot (v.1.22.0) was used to visualize the results [94]. For all analyses, the *p*-value was adjusted using the Benjamini–Hochberg method to control the false discovery rate (FDR). Categories with a cutoff of *p*. adj < 0.05 were considered significant.

### 4.9. Deconvolution

NuSVR deconvolution was performed using the Tabula Sapiens v1 basis matrix (https://github.com/sevahn/deconvolution (accessed on 1 March 2024)) [95]. As recommended by the authors, data were CPM normalized and no log transformation was performed.

### 4.10. Statistical Analysis

Analyses and graphs were produced using Prism (Graph Pad, version 10) or the R statistical environment (v.4.0.5) [96]. Differences in tumor volumes or immune cells were compared using a Welch and Brown–Forsythe ANOVA followed by Tukey’s multiple comparisons test with significance levels set at 0.05. Error bars represent the standard error of the mean (SEM). Mice sacrificed because of tumor volumes > 1500 mm^3^ or weight loss >20% retained this value until the end of the experiment to avoid incorrect improvement of group means due to dropouts. Kruskal–Wallis tests were used to compare cfRNA concentration between multiple groups, and Wilcoxon rank-sum tests were used to compare between two groups. For individual testing, *p*-values smaller than 0.05 were considered significant. In cases of multiple testing, the Benjamini–Hochberg procedure was used to calculate false discovery rate adjusted *p*-values (*q*-values), and significance was defined as *q*-values smaller than 0.05. Binary logistic regression, R glm function with binomial family, was used to classify samples using a training and test set. Based on these predictions, the Receiver Operating Characteristic (ROC) curve and Area Under Curve (AUC) were calculated and visualized using pROC (v.1.18.0) [40]. Correlation analysis was carried out using the Spearman correlation coefficient (R) between different metrics.

## Figures and Tables

**Figure 1 ijms-25-09982-f001:**
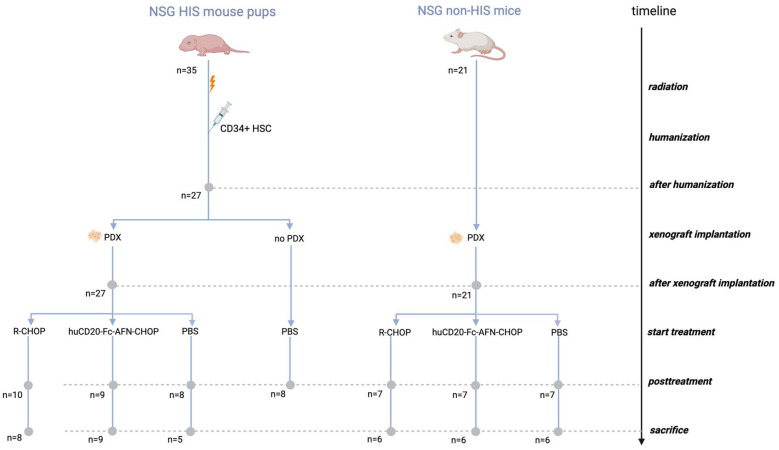
Overview of blood plasma samples in the study derived from both HIS and non-HIS NSG mice. The gray dots in the figure represent a blood draw by tail vein puncture. HIS: human immune system; NSG: NOD-scid IL2Rgnull; PBS: phosphate-buffered saline; PDX: patient-derived xenograft; NSG: NOD-scid IL2Rgnull.

**Figure 2 ijms-25-09982-f002:**
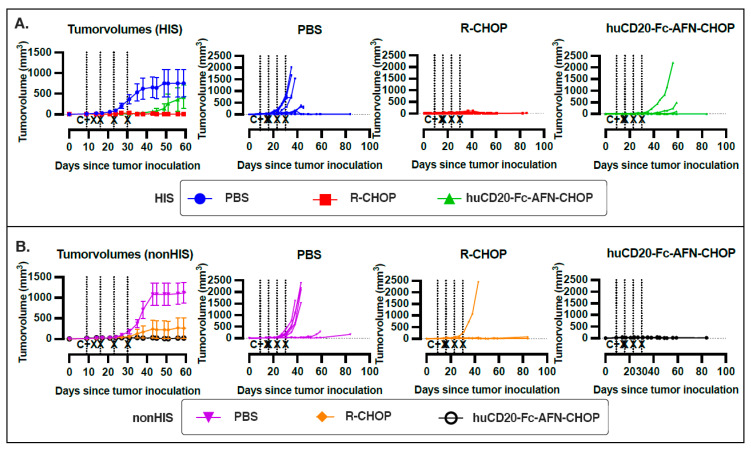
Treatment response. Tumor volumes from the start of tumor inoculation in HIS (**A**) and non-HIS (**B**) NSG mice. For the grouped analyses in the first graph of each panel, the tumor volume of sacrificed mice was kept at 1500 mm^3^ to avoid inappropriate improvement of group averages. For the other graphs, each line represents an individual mouse. Vertical lines indicate IV treatments (C = CHOP; X = Rituximab/huCD20-Fc-AFN/PBS). Error bars represent the standard error of the mean (SEM). HIS: human immune system; NSG: NOD-scid IL2Rgnull; PBS: phosphate-buffered saline.

**Figure 3 ijms-25-09982-f003:**
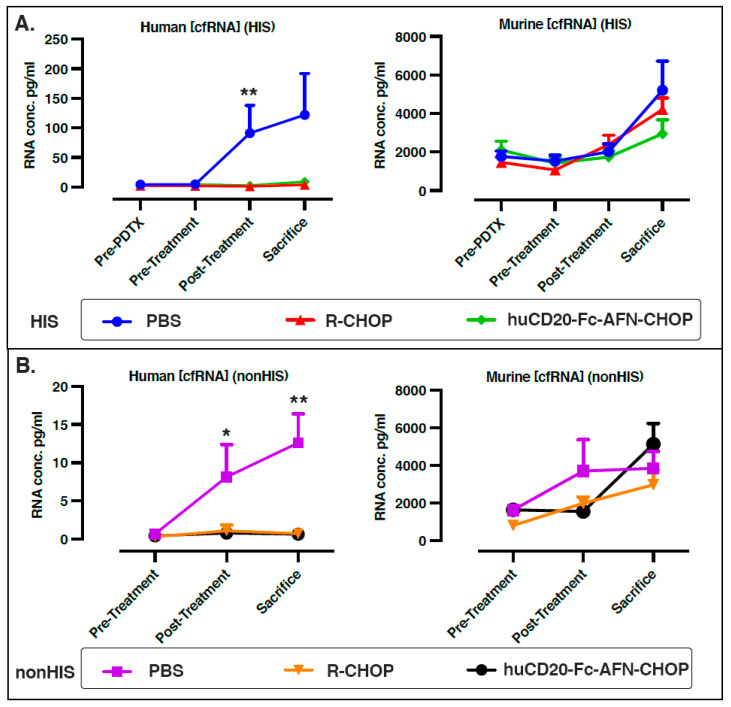
Endogenous human and murine cfRNA concentrations in HIS (**A**) and non-HIS (**B**) PDTX NSG mice treated with PBS, R-CHOP, and hu-CD20-FC-AFN-CHOP. Significant *p*-values are shown. CfRNA: cell-free RNA; HIS: human immune system; NSG: NOD-scid IL2Rgnull; PBS: phosphate-buffered saline; PDTX: patient-derived tumor xenograft. Pre-PDTX: before xenograft implantation; * *p*-value < 0.05; ** *p*-value < 0.01.

**Figure 4 ijms-25-09982-f004:**
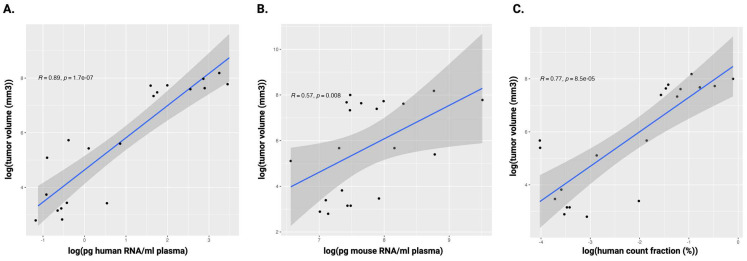
Spearman rank correlation between tumor volume (mm^3^) of non-HIS xenografted PBS-treated NSG mice and human ctRNA concentration (**A**), murine cfRNA concentration (**B**), and the fraction of tumor-derived counts over all endogenous counts (**C**). Log-transformed values are shown. The darker gray region represents the range where the true regression line could lie with 95% confidence. ctRNA: circulating tumor RNA; HIS: human immune system; NSG: NOD-scid IL2Rgnull; PBS: phosphate-buffered saline; PDTX: patient-derived tumor xenograft; R: Spearman correlation coefficient.

**Figure 5 ijms-25-09982-f005:**
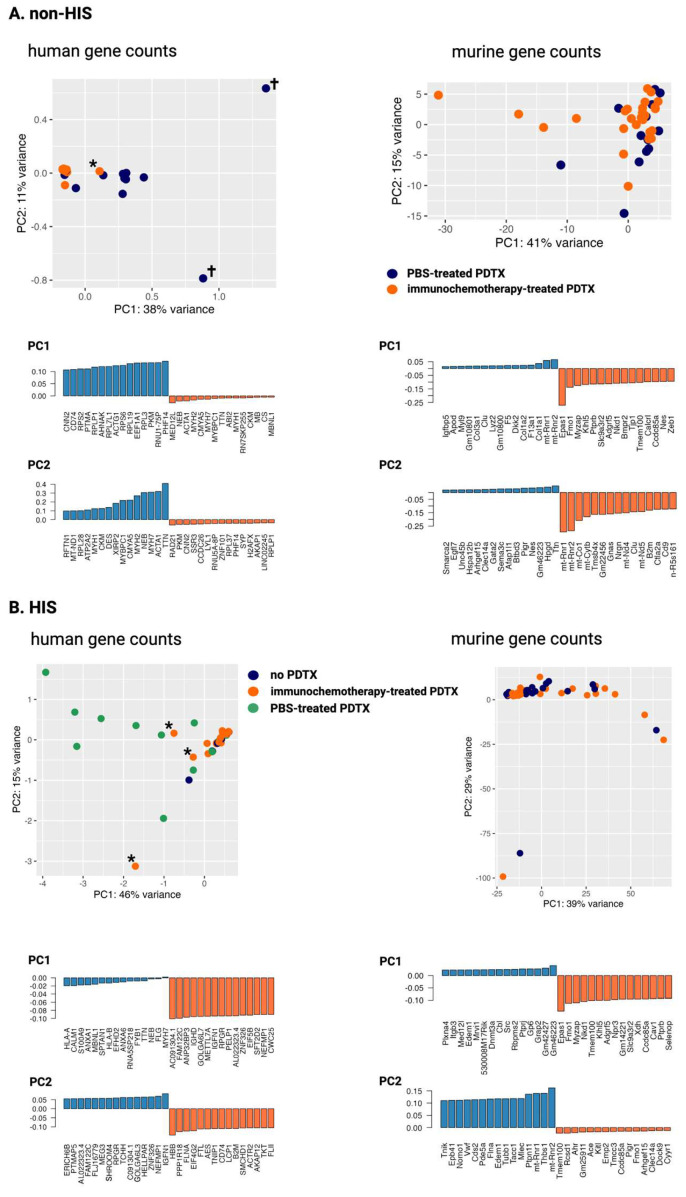
Principal component analysis for both human- and murine-normalized counts within the non-HIS (**A**) and the HIS (**B**) NSG mice. The main drivers of variance of the first two components are shown for each of the comparisons. The symbol † indicates the PBS-treated samples with the highest tumor volumes. The symbol * indicates samples with tumor growth despite immunochemotherapy. HIS: human immune system; NSG: NOD-scid IL2Rgnull; PDTX: patient-derived xenograft model; PBS: phosphate buffered saline, PC: principal component.

**Figure 6 ijms-25-09982-f006:**
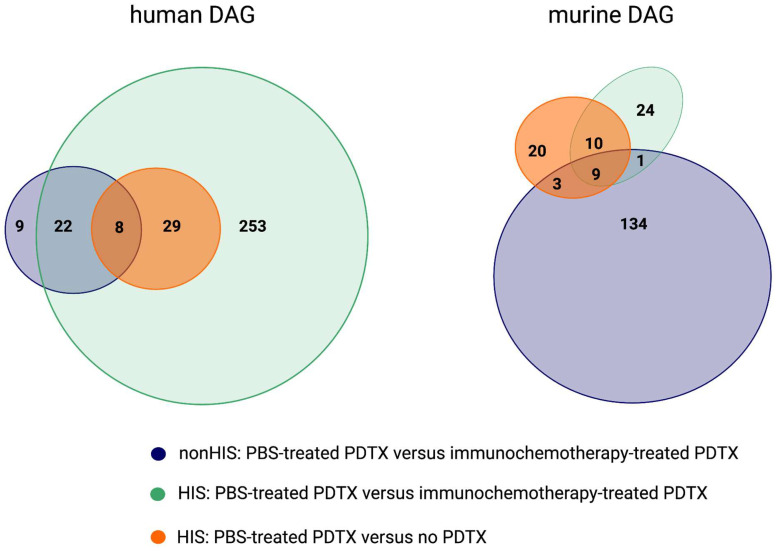
Venn diagram illustrating the overlap of differentially abundant human and murine genes between the blood plasma of different subgroups. DAGs: differentially abundant genes; HIS: human immune system; PBS: phosphate-buffered saline; PDTX: patient-derived tumor xenograft.

**Figure 7 ijms-25-09982-f007:**
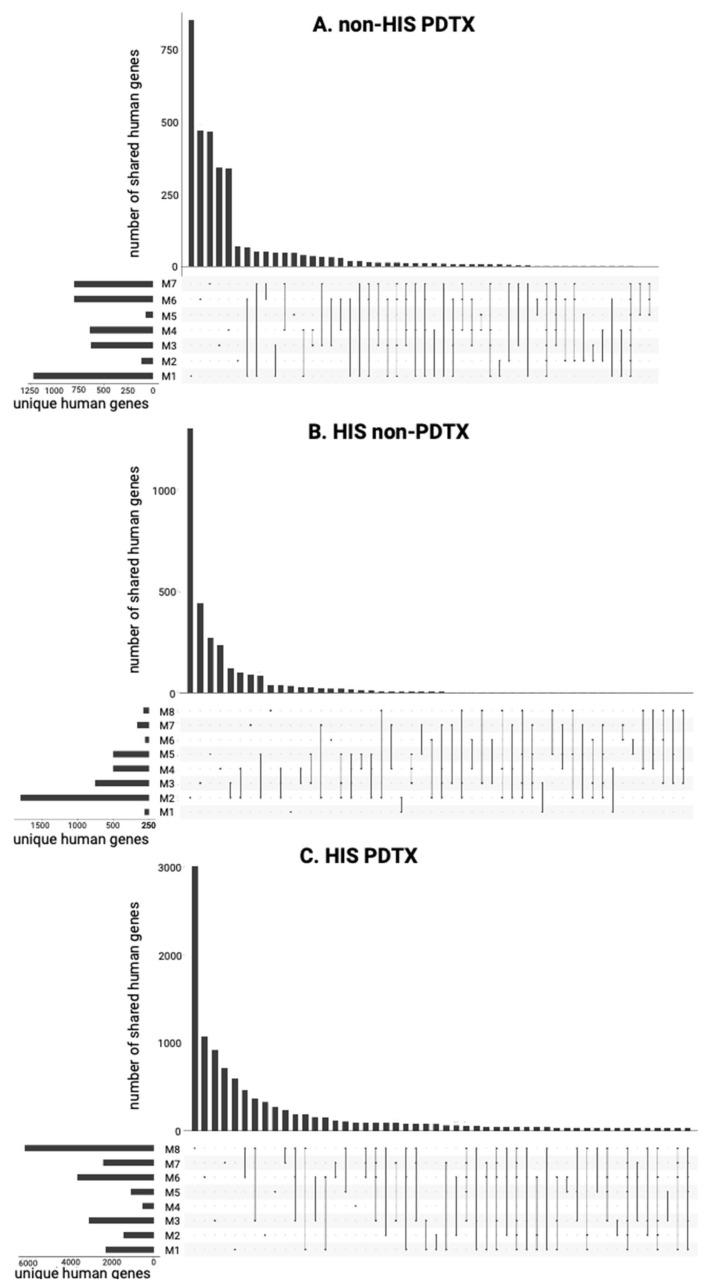
Overlap analysis of unique human genes among mice of the non-HIS PDTX (**A**), HIS non-PDTX (**B**), and HIS PDTX groups (**C**). For each group, the top section of each panel shows the number of unique genes for each mouse or combination of mice, the lower left section depicts the number of unique human genes per mouse, and the lower right section indicates which mice (as indicated by the dots) share unique genes. HIS: humanized; PDTX: patient-derived tumor xenograft.

**Figure 8 ijms-25-09982-f008:**
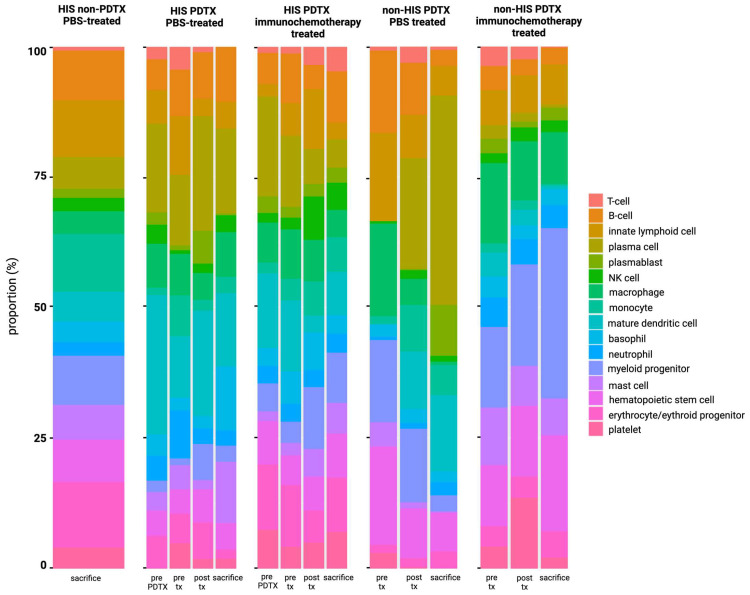
Computational deconvolution results on human cfRNA derived from blood plasma demonstrating the relative contribution of each cell type to the circulating immune system, red blood cells and platelets during xenograft growth and treatment in PBS-treated HIS non-PDTX, PBS-treated HIS PDTX, immunochemotherapy-treated HIS PDTX, PBS-treated non-HIS PDTX, and immunochemotherapy-treated non-HIS PDTX NSG mice. CfRNA: cell-free RNA; HIS: human immune system; NSG: NOD-scid IL2Rgnull; PBS: phosphate-buffered saline; pre-tx: pre-treatment; post-tx: post-treatment.

## Data Availability

Raw RNA-sequencing data are being deposited in the European Genome-phenome Archive (EGA; dataset EGAD50000000806).

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
