# Peer review of "Characterizing the Cell-Free Transcriptome in a Humanized Diffuse Large B-Cell Lymphoma Patient-Derived Tumor Xenograft Model for RNA-Based Liquid Biopsy in a Preclinical Setting"

_ijms, 2024, doi:10.3390/ijms25189982_

Round 1

Reviewer 1 Report (Previous Reviewer 1)

Comments and Suggestions for Authors

The authors have amended the manuscript based on suggestions from the reviewers and the paper is now suitable for publication.

Author Response

We would like to thank the reviewer for the thorough revision of our manuscript. 

Reviewer 2 Report (New Reviewer)

Comments and Suggestions for Authors

 This article has left me with mixed feelings. On the one hand, the research team's meticulous study design is commendable, the methodology is correct, and conclusions are justified by the results. However, the presentation quality is regrettably deficient, suggesting a lack of consideration for the readers' comprehension.

This is a good quality and potentially quite important study, and it deserves a better presentation. I've attached the pdf file with a lot of comments that I hope will help the authors improve the readability of their paper, as well as some minor remarks.

Author Response

This manuscript is a resubmission of an earlier submission. The following is a list of the peer review reports and author responses from that submission.

Round 1

Reviewer 1 Report

Comments and Suggestions for Authors

Ducruyenaere et al. write an interesting report performing RNA-based liquid biopsy of humanized DLBCL pateint-derived tumor xenograft mice. By distinguishing human from murine sequencing reads in RNA present in murine plasma by a highly accurate combined mapping strategy, the effects of humanization, xenografting and their interplay were assessed. The following points should be addressed before consideration for publication.

1. In Figure 1, speculate why R-CHOP was more effective in controlling tumors than huCD20-Fc-AFN-CHOP in humanized mice.

2. AFN (probably "AcTaferons"?) needs to be spelled out when first used and added to the abbervations list.

Reviewer 2 Report

Comments and Suggestions for Authors

This article is about characterization of the cell-free RNA in the blood in a context of DLBCL. The issue of this article is of high interest considering the spread of studies on the value of circulating RNAs as cancer biomarkers and the concomitant difficulty in confirming their tumor origin. The experimental design of this study, that include both humanized and non-humanized PDX models, allows the dissection of the origin of cfRNAs in the blood.

However, some aspects of the study need a clarification on order to improve the quality of the manuscript:

1) At page 5 (line 168) the concept of “healthy murine cfRNA” is not clear, in my opinion, since it is referred to PDX mice, so they cannot be considered healthy.

2) Considering the human DAGs, they appear to be all increased in the blood comparing non-treated progressing mice and treated “cured” mice. However, when considering the 8 tumor-specific gene signature, in figure S11 some of these transcripts are shown to decrease with tumor progression in the human patient, as if in a human context there isn’t the same correlation between tumor progression and the quantity of these RNAs in the blood. Authors should discuss this result since it is fundamental to assess the applicability of these results to a human context. Furthermore, since these results are about a single patient, it would be interesting to validate the data in the blood using some larger patient dataset.

3) Since the signature of 8 gene is able to predict the therapy response in the PDX models, it would be interesting to assess its predictive value also in a human context (while only the diagnostic value is considered in the article, since comparison between DLBCL and healthy subjects is performed).

4) Figure 6 is of difficult interpretation since it is not clear what the X axis is referred to.

5) Regarding the human samples used in this study, there should be more details about the cohort of patients used for the data from the FFPE tumors (I suggest a table with the biological and clinical features of patients). Furthermore, some details about the time point of blood collection from the human patient should be added in Materials and Methods.

6) In my opinion it would be interesting to discuss the fact that some RNA species (e.g. the mitochondrial RNAs and the microRNAs) are only present among the DAGs found in the HIS models and absent from the DAGs found in the non-HIS models.

7) As minor concerns:

-please note that in the Figure 3 the letters of the panels are not reported

-the use of the HIS abbreviation is confounding since it stands for "human immune system" but is also used for humanized mice.